# Dynamic Data Feeding into BIM for Facility Management: A Prototype Application to a University Building

**Jonatan Villavicencio Moreno** [1], **Rita Machete** [2], **Ana Paula Falcão** [2], **Alexandre B. Gonçalves** [2] **and Rita Bento** [2,*]

1    Instituto Superior Técnico, Universidade de Lisboa, Av. Rovisco Pais, 1049-001 Lisboa, Portugal; jonatan.moreno@tecnico.ulisboa.pt
2    CERIS, Instituto Superior Técnico, Universidade de Lisboa, Av. Rovisco Pais, 1049-001 Lisboa, Portugal; rita.f.machete@tecnico.ulisboa.pt (R.M.); ana.p.falcao@tecnico.ulisboa.pt (A.P.F.); alexandre.goncalves@tecnico.ulisboa.pt (A.B.G.)
\*    Correspondence: rita.bento@tecnico.ulisboa.pt

**Abstract:** Building information modelling (BIM) has demonstrated its potential as a solution providing support to a series of operations related to facility management (FM) through building data retrieval, analysis, and processing. However, some challenges to the effective adoption of BIM-centred FM information systems occur in their design and implementation, causing obstacles to usability. Among these challenges are the customization of the information structure for each application case, the dynamic character of data supporting building maintenance, and the range of FM specialities involved, frequently including persons who are not BIM experts. This paper presents a BIM–FM prototype to support operations and access updated environmental data for a university building. The two contributions of the developed prototype are its ability to register two types of dynamic data, namely, the regularly acquired environmental sensor information and the sporadic building intervention records, and the automation of the data feeding, updating, and retrieval processes, allowing a user-friendly environment for both BIM experts and non-BIM users. Exploring the BIM interoperability and the integration of plug-ins, the proposed solution enables the comprehensive registration of dynamic FM-related data in an updated model while being accessible to all the specialities involved in the building management operations, enhancing its usability as an integrated solution for data maintenance and retrieval.

**Keywords:** building information modelling; facility management; university building; wireless sensor; FM–BIM integration





## 1. Introduction

Building information modelling (BIM) has emerged as a collaborative paradigm for adequate and coherent building information storage and sharing between the various professionals involved in the architecture, engineering, and construction (AEC) industry. As a global solution for handling building data, BIM allows a 3D graphic representation, together with a database for attribute registration. Interoperability is critical to allow sharing, reporting, and updating the information contained in building models, using various data formats and programs and involving the diverse professionals and stakeholders related to the building life cycle [1]. The modelling of existing buildings has been an important issue since the emergence of BIM in the AEC industry. In the case of existing buildings, BIM integration has been framed as a conceptual diagram with nested levels, from the 3D geometric representation to the use of 7D BIM for preventative maintenance scheduling or disaster and emergency planning [2].

Facility management (FM) concerns the phase of operation and maintenance within the building life cycle. In FM, building data in BIM can be used in daily management operations such as obtaining digital or printed layouts, charts, or symbolic representations

of the status of the building. This has applications in many areas covered by FM, such as the management of energy consumption, security of facilities, and the tracking of maintenance and repair works [3].

The integration of BIM and FM (BIM–FM) into solutions supporting building managers is not without a series of issues and challenges. A comprehensive review of such issues is provided by Dixit et al. [4]: the authors identified problems ranging from individual lack of awareness of the integration benefits and low motivation in BIM adoption to more technical issues such as the low level of interoperability between authoring tools for BIM and FM.

One of the cases where this integration seems promising is in university buildings. These are generally multipurpose constructions involving a diverse range of activities, from classrooms to offices, laboratories, services, and technical and facility support spaces. From a management point of view, this generates a diversity of features and attributes (e.g., areas and furniture for classrooms, equipment for laboratories, etc.), usually with a large number of stakeholders involved in daily management activities (e.g., classroom spaces managed by a central university office, laboratories by research groups, offices by a department, etc.). There is, therefore, a subsequent multiplication of data sources and platforms related to the diverse building subsystems. Information of interest for managers is also usually dense and constantly changing, as university buildings operate year-round, sometimes with spaces open 24/7, and are used by a large university community, thus requiring constant maintenance and access to updated databases. As a consequence, a centralized solution for the management of complex buildings, enabling access for energy, comfort, safety evaluation, and monitoring purposes, would be a valuable asset for managers.

BIM, as a centralized repository with strong capacities for 3D visualization, has emerged as an interesting solution, with some examples of exploratory BIM–FM integration in university buildings in the literature. However, the dynamic character of facility maintenance data of interest for BIM-based solutions has not been comprehensively addressed, and this constitutes a lack in BIM–FM research. These data can be continuous, e.g., from sensors placed for energy and comfort monitoring and assessment, or intermittent, e.g., concerning management operations such as periodic interventions that need to be registered.

This paper aims to foster the implementation of BIM–FM by presenting and discussing a solution for the automation of BIM data updates from distinct data feeding sources, within the university building context. As such, its goal is to explore the integration of several types of data into the information update cycle of a BIM model applied in facility management. The implemented solution enables the automation of data feeding and enhances the centrality of BIM as a platform to maintain and provide updated information supporting FM activities. The case study is the Civil Engineering Pavilion, a multipurpose university building located on the Alameda campus of the Instituto Superior Técnico, University of Lisbon, in Portugal. The prototype deals with data originating from environmental sensors, as well as alphanumeric data from the daily maintenance records kept by the building manager, covering information with both continuous and irregular update periodicities. It also provides access to meaningful information for the diverse areas of expertise and non-BIM-expert technicians who require the retrieval of building data.

The paper is organized as follows: a literature review concerning the application of BIM-centred FM in university contexts is provided in the Background section. The workflow is described in the General Methodology section, and then the Case Study section details the integration of dynamic data within the developed model. After the Results section, focusing on the outcomes of the implemented solution, the paper includes Discussion and Conclusions sections and the implications for future research.

## 2. Background

### 2.1. Concepts

Requirements for BIM in the facility management domain are different from those oriented towards other stages of the building life cycle [5]. During the project and construction phases, BIM is generally used to support the diverse activities of professionals and experts dealing with cost estimation, the optimization of resources, and the sequential programming of tasks. In contrast, during a building's operational stage, the use of BIM–FM is more demanding, with many attributes with frequent value changes for which is important to access their historical record, a high level of attribute heterogeneity and formats, and the need to make the information accessible for those who are not BIM experts [6]. Despite these difficulties, the advantages of the use of BIM in the scope of building management as a centralized database for all data about the building, supporting maintenance, and management, have been identified in the literature for many aspects of the building in operation, ranging from safety and health [7] to energy consumption [8]. These applications are in general highly demanding in terms of attribute updating because, in the context of decision support, it is not only important to access the current status of the building but also the registry of its past. As an example, the analysis of the energy consumption of a room can be supported by a series of relevant dynamic parameters such as temperature and luminosity, combined with the attributes of that space and its environmental factors such as its location in the building, dimensions, or materials.

The use of embedded intelligence in buildings, known as smart building, is empowered by the decrease in the price of digital sensors and the computer power to handle streams of big data. Wireless sensor network (WSN) technology has gained prominence within the management of built environmental assets post-construction, linked to BIM approaches ([9–11]). A WSN incorporates a wireless network of spatially distributed autonomous devices or sensors, to provide real-time monitoring of physical or environmental conditions. Most recent buildings nowadays are equipped with indoor environmental sensors, mainly for collection protection [12,13] or for energy sustainability assessment and analysis of comfort performance [14]. Environmental and acoustic sensors can provide valuable information concerning the building occupancy [15–18] and the environmental conditions to which the building is subjected in real time [19]. Nevertheless, the automation of data evaluation has yet to be properly implemented, due to challenges such as the size of datasets, levels of detail, and interoperability with existing formats employed [20]. With regard to the integration of BIM and the Internet of Things (IoT), most studies are mainly theoretical and conceptual [21–23] or oriented towards prototype solutions [24–26].

### 2.2. Applications to University Campus Buildings

The preservation and safeguarding of campus buildings is a permanent concern of universities. It is therefore of the utmost importance to have resources and tools to promote the building's safety and sustainability. Indeed, some major challenges arise in the case of university buildings using sensors within a BIM-based facility management process, such as the difficulties in installing the sensors in a non-invasive way when there is a heritage value assigned to the constructions, or the difficulty in ensuring the integration, in real time of data gathered by the sensors in a 3D platform, e.g., environmental and acoustic sensors in highly populated and intensively used buildings.

A theoretical framework for BIM–FM integration in the scope of university buildings' operations and maintenance, was suggested by Galiano-Garrigós and Andújar-Montoya [27]: the research highlighted the importance of adopting BIM together with FM to provide more efficient maintenance using emerging technologies for data capture, such as smart sensors. A case study on the applicability of the proposed paradigm to a building within the University of Alicante campus demonstrated its feasibility and potential as a comprehensive system for accessing and sharing information between the various stakeholders involved in building maintenance.

A second example deals with a total of 32 non-residential buildings in the Northumbria University campus in England, that were modelled using BIM–FM [28]. The study enabled an evaluation of the value of such solutions, specifically the improvement offered by the FM data accuracy when compared to traditional information handover processes and the increase in the efficiency of work-order execution. It also enabled an enumeration of some of the major challenges in the process, such as issues related to interoperability and the presence of several operating systems for managing the buildings and the lack of specifications related to BIM for FM modelling requirements.

One other example of the type was developed at the Department of Civil Engineering at the National Taipei University of Technology [29]: this study proposes a system to discuss and analyse the entire maintenance management process. The model focused on facilities maintenance work, and a colour code was assigned to the BIM elements to enable easy understanding of the status (complete/under work, delayed/not delayed, etc.) of maintenance works. A comparison of the presentation, recording, and searching of information between the BIM solution and the traditional paper-based and management information system is also included.

Muñoz Pavón et al. [30] describe the extent to which the digitization of information has impacts on the university buildings' maintenance and management. The differences between traditional campuses, e-campuses, and digital campuses are also detailed in terms of information processing, highlighting, in the last case, the role of technologies such as radio-frequency identification (RFID), the Internet of Things (IoT), and sensor technology, among other emerging technologies [31]. Important economic benefits arise from the integration of such technologies as data providers for campus buildings' management subsystems for energy and water consumption, waste handling, and staff administration, amongst other things. This integration can be supported by a BIM solution, with its additional benefits of 3D visualization and spatial interpretation.

A sensor–BIM integrated solution ([25]) for a pavilion in the University of Cagliari used a data platform to access several measured indoor building conditions such as temperature, luminance, and energy consumption parameters, via a BIM interface (Revit). The integration of this model and the data from sensors is based on Dynamo visual programming, populating the BIM with sensor-acquired data. The goal of the solution was to provide the means to assess the comfort and energy efficiency of the building, facilitating energy audit procedures.

Villa et al. ([32]) presented a fully automated framework composed of IoT sensors and a BIM platform, to support preventive maintenance in a laboratory room at the Politecnico di Torino. The integrated IoT and BIM architecture has shown its applicability not only for preventing anomalies but also for facilitating communication with building managers, providing support for better decisions with real-time data.

## 3. General Methodology

The proposed workflow is adapted from the guidelines suggested by McArthur ([33]), setting the foundations of the model in identifying the needs of the client and forecasting the data that will enrich the 3D model. The stages of the workflow are described below (Figure 1).

**Identification of FM Processes**. Identifying the current facility management and maintenance workflow is mandatory for determining the scope and the structure of the information of the future building model. With the help of the local maintenance team, the outcomes in this stage must be the scope of the project, its functional requirements, and the levels of detail of the 3D model and data that are compatible with such requirements. It is also necessary to know the current protocol for registering, organizing, and accessing information concerning operation and maintenance. This includes a survey of the available and intended data sources and formats, the frequencies of data updating, the user profiles in terms of data access for all the involved technicians, and the relations between their

activities. Based on this, a general functional schema for the information flow in the system must be defined.

| **Identification of FM Processes** | **Spatial Data Collection and Attribute Definition** | **Development of the system** | **Model Data Feeding** |
|---|---|---|---|
| • Scope and the structure of the information of the future building model.<br>• Current protocol to register, organize and access the information concerning operation and maintenance. | • Survey the geometric data.<br>• Data structure for the attributes linked to the identified spaces - **Attribute Tree**. | • **Systemic solution**:<br>- Representing the 3D geometry and the attributes defined previously;<br>- Acquisition, storage, and migration of dynamic data linked to the BIM. | • BIM reflects information such as the current building status, while the system keeps the capacity to access previous registries of its inventory. |

**Figure 1.** Workflow adopted for BIM–FM development.

**Spatial Data Collection and Attribute Definition**. The importance of the techniques used to survey the geometric data lies in the level of detail required for the project. Existing plans and/or dedicated acquisition of spatial data, e.g., using terrestrial laser scanning complemented with in situ measurements, can support the 3D construction. Additionally, an adequate data structure for the attributes linked to the identified spaces must be established, following the context decided at the previous stage. This results in an inventory of spatial data sources and an attribute tree that will be the bases for the development of the building model.

**Development of the system**. The implementation of BIM implies the selection of software that is capable of representing the 3D geometry and the attributes defined previously. This involves the definition of BIM objects, which can be principal elements such as walls, ceilings, and floors, or connecting elements such as windows and doors, that will define the spatial management unit used in the FM process. Additional software to handle the acquisition, storage, and migration of dynamic data linked to the BIM may be chosen for its capacity for allowing data exchange with the BIM model. Specifications for application programming interfaces (APIs) are particularly useful for ensuring the correct data flows between the system components, accessing relevant information, and updating the BIM-centred system.

The deliverable at this stage is a systemic solution that includes a BIM model based on the available geometric information with a predefined level of complexity and following the defined attribute specifications, together with additional programs that connect with the BIM under an API, implementing the functional requirements defined previously.

**Model Data Feeding**. Specifically for FM purposes, it is important that the BIM model reflects information such as the current building status while the system retains the capacity to access previous registries of its inventory. Depending on aspects such as the acquisition mode and the frequency of updating, data feeding into the system can be performed either automatically or manually. The model geometry and attributes can be edited directly in the BIM software, an activity that is usually performed by BIM experts. However, the attributes may also be modified through the established API, where users who are not BIM experts can edit the information stored in the data sources connected to the BIM model. Through the activation of a data updating tool, the attributes of BIM elements are then changed accordingly.

## 4. Case Study

The Civil Engineering Pavilion at Instituto Superior Técnico (IST) is on the campus of Alameda in Lisbon, Portugal (Figure 2). This building was designed in 1982 and built in the 1990s. It has 403 rooms, including laboratories, classrooms, and staff rooms, and it occupies a former garden area in the northwest corner of the campus, on a sloped parcel. It has seven stories occupied by classrooms, offices, an auditorium, a museum, a library, a bar, a canteen, terraces, laboratories, technical galleries, and a garage. The building floor space is ca. 100 m by 55 m.

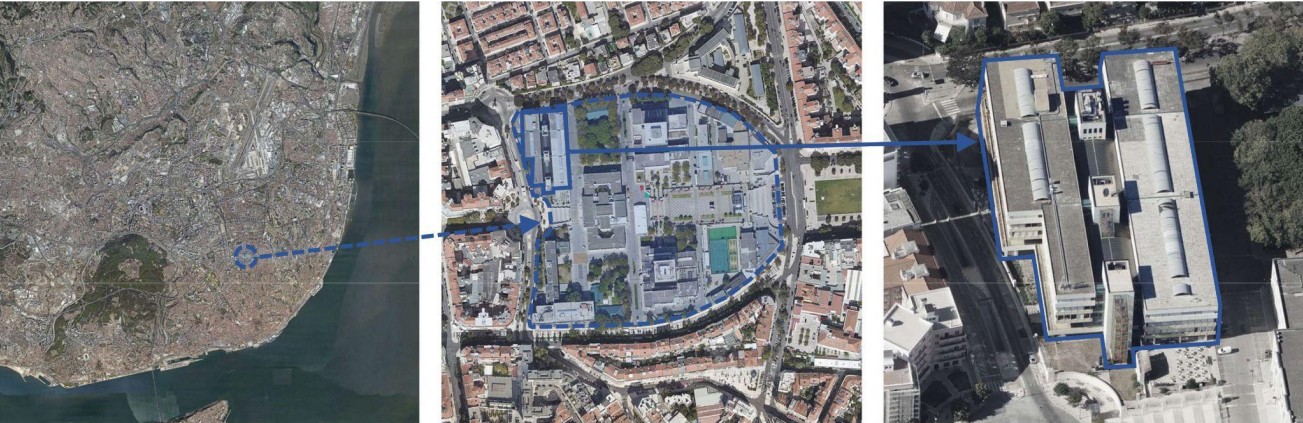

**Figure 2.** Location of the case study: Lisbon city (**left**), IST campus (**centre**), and Civil Engineering building (**right**).

The supervision of the building is directed by a facility manager, who is responsible for handling various types of external requests from other campus services such as security and major works of maintenance, while keeping an updated inventory of the building spaces. The facility manager also fills in forms requesting interventions from the external services, using a ticket-based process. A centralized system, common to several buildings within the campus, keeps the information concerning these tickets in a relational database with intranet access but without graphical representation of the locations relating to the information. In addition, the facility manager deals with requests that need to be addressed and are not covered by the above-mentioned system, mainly relating to minor interventions that are managed by allocated building maintenance staff. The manager keeps ad hoc track of the requests, without any formal registration of the events. The proposed BIM-based solution will focus on the latter information processes, as the campus-scale system for the registration of major interventions should be retained.

### 4.1. Identification of FM Processes

Meetings with the building manager enabled identification of the flows of information involved in the management process and its agents. Both management activities that require registration in the central campus system (e.g., interventions in the power network or major building works), and those that are decided or organized locally by the building manager, (e.g., door locks administration, cleaning, or maximum room occupancy setting) were inventoried. The resulting functional schema enabled the scope of the building model to be delimited (Figure 3).

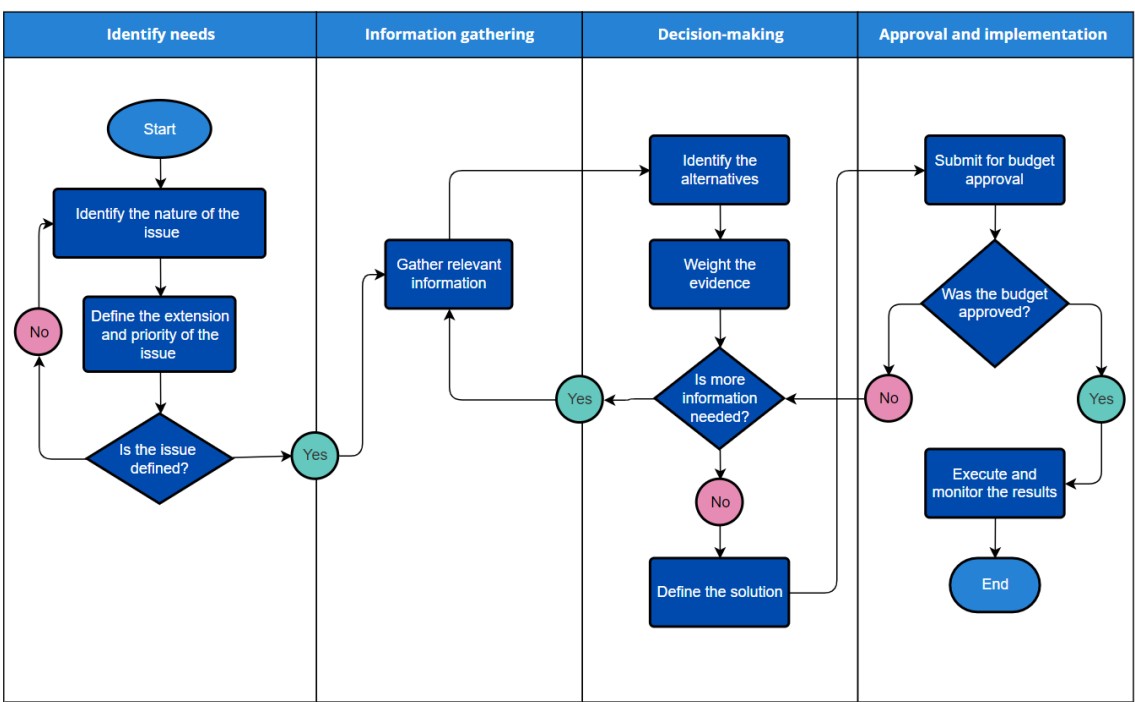

**Figure 3.** Functional schema resulting from the meetings with the building manager.

### 4.2. Spatial Data Collection and Attribute Definition

A set of CAD drawings with floor plans, one for each level of the building, was provided by the building manager and used as the main source for the spatial model. This was complemented by local visual inspection and measurements, such as the heights of the seven building levels and the dimensions of windows, due to the lack of façade drawings. This was possible as the building geometry is very regular, enabling later parametrization of the process. As the graphical representation dates from the construction of the building, validation of the internal spatial disposition (such as splitting and merging rooms) was performed.

The information deemed to be relevant for associating with the graphical representation of the building was arranged into four groups: "space managing", "maintenance data", "asset conditions", and "sensor data". These information groups can be classified into two categories according to their frequency of change: the three first groups consist of data that are changed sporadically, referred to herein as "occasional data", while data from sensors are referred to as "continuous data".

Under the "space managing" group, the selected attributes were:

- Occupation limit—denoting the maximum number of people that can be inside a specific room;
- Room uses—describing the specific use of a room in the model (e.g., "laboratory", "classroom");
- Net area—displaying the surface area of the room;
- Department info—denoting the subgroups within the department managing the room (e.g., Architecture, Structures, Mining and Georesources, Construction);
- Room availability—indicating whether a room is available or unavailable;
- COVID-19 occupation—denoting the number of persons allowed in the room following the social distancing rules imposed during the pandemic period.

Under the "maintenance data" group, the selected attributes were related to technical information such as links for accessing the campus internal maintenance databases, main-

tenance frequency and data related to service life, and a registry of the successive works of maintenance:

- Technical information—containing external links to a shared folder with datasheets and technical information related to assets;
- Life cycle—indicating the estimated service (time units) life of the assets according to the manufacturer;
- Intervention—containing information about the last (or ongoing) intervention assigned to a room or asset.

Under the "asset conditions" hierarchy, the selected attributes were related to facilities controlled by the manager:

- HVAC data—relating to the HVAC condition, classified as "operating", "not operating", or "not applicable";
- Window blinds—referring to the window blinds condition, classified as "operating", "not operating", or "not applicable";
- Lock type—storing data for locks and keys, which are of different types.

Under the "sensor data" group, environmental data are stored, displaying the last values registered by sensors and presenting a link for accessing historical data for each variable.

- Illuminance (lx)—indoor light level in lux;
- Pressure (hPa)—indoor air pressure value in hectopascals;
- Relative humidity (%)—value of relative indoor humidity as a percentage;
- Temperature (°C)—value of indoor temperature in degrees Celsius.

Floors, walls, windows, doors, and ceilings were selected as the principal elements that enabled the definition and registration of spaces (named *rooms*), in line with the spatial unit for building management. As such, data for the "space managing", "maintenance data", and "asset conditions" groups were assignable to rooms, while the data in the "sensor data" group were associated with walls, since the location of sensors inside a room influences the values that are registered. These locations were defined by a specific position in the wall, within a room.

The corresponding attribute tree was established, as illustrated in Figure 4.

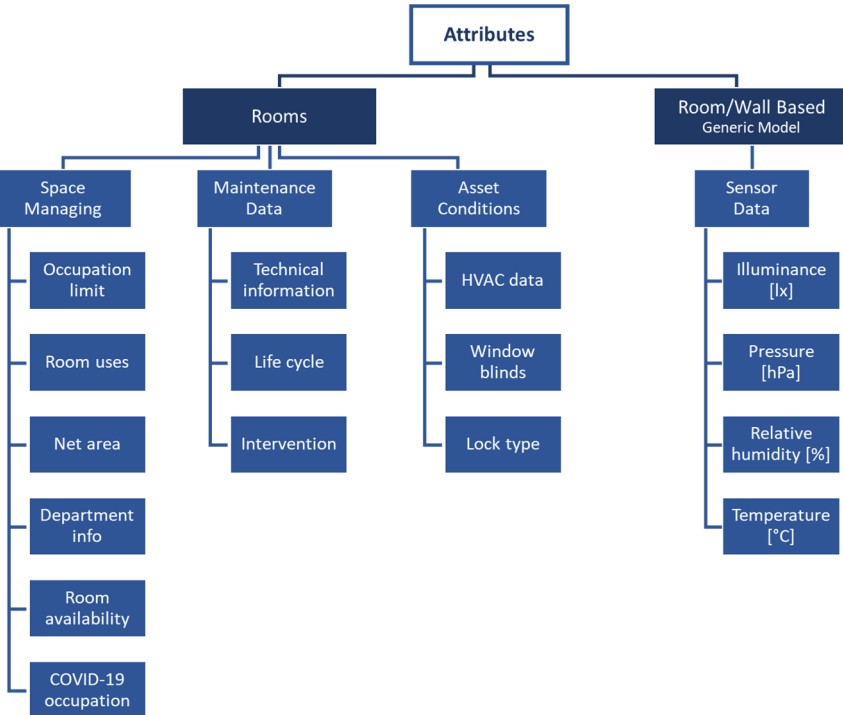

**Figure 4.** Tree of attributes representing the data associated with the model.

### 4.3. Development of the System

A building model was developed using Autodesk Revit®, a general BIM software tool that is well known in the AEC industry, with interesting capacities regarding interoperability.

A generalized level of detail, with a 3D geometry that described the approximate size, shape, and location, was applied to enable both a clear space description and a comprehensive geometric representation of the building, while keeping the model light and usable for maintenance purposes (Figure 5).

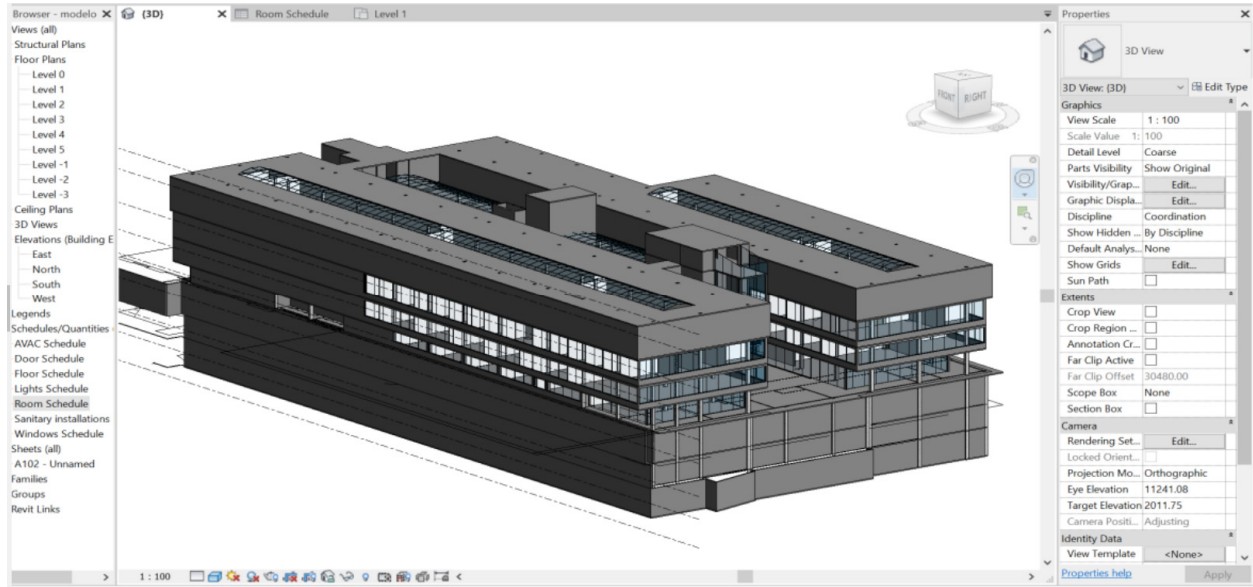

**Figure 5.** Autodesk Revit® model of IST's Civil Engineering Pavilion.

The BIM elements corresponding to each of the 403 identified rooms were defined by selecting the appropriate walls, floors, ceilings, columns, windows, and doors (for each of these instances, free-text information concerning the type of material was included in the model).

The system also included spreadsheets to register the historical values of attributes, enabling this information to be consulted through resource locator links stored as attributes of the BIM objects. Spreadsheets were also used to store the sensor measurements. Upon completion, BIM acted as the central platform for data storage, enabling the extraction of element information, maintaining an updated version of attributes through a dynamic connection to external spreadsheets, and facilitating the updating of attribute data in the Autodesk Revit® model by users who are not BIM experts.

For these purposes, and for supporting the interaction with the building model, an interface was developed to work as a bridge between the different tools used (Figure 6).

Schedules with editable information on the rooms and assets were exported to the Microsoft Excel® worksheet format (*.xlsx) via the DiRoots® plug-in for Autodesk Revit® [34]. Then, after editing the Excel spreadsheet, the same plug-in was used to update the BIM model with the edited information. In parallel, responding to the need to have a historical record of interventions, a VBA script enabled all data to be saved and registered in a different workbook, allowing the dataset to be consulted and filtered using regular Microsoft Excel® functions. Regarding data management, the BIM database was edited not directly using Revit® but using the spreadsheet that was linked to the model. Data were updated only after verification, conducted by the manager, with respect to the content and quality of the input data in reports (Figure 7).

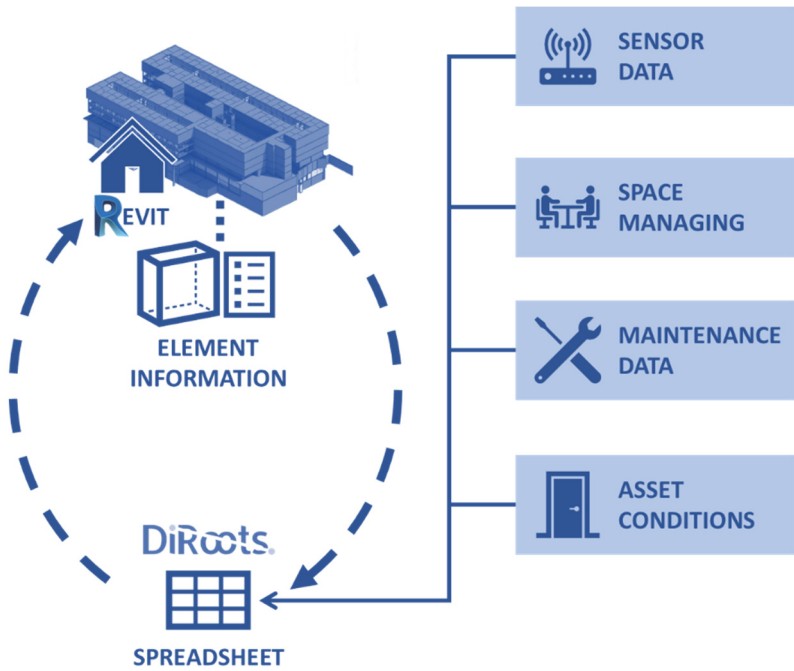

**Figure 6.** The interface for BIM data updates using the DiRoots plug-in.

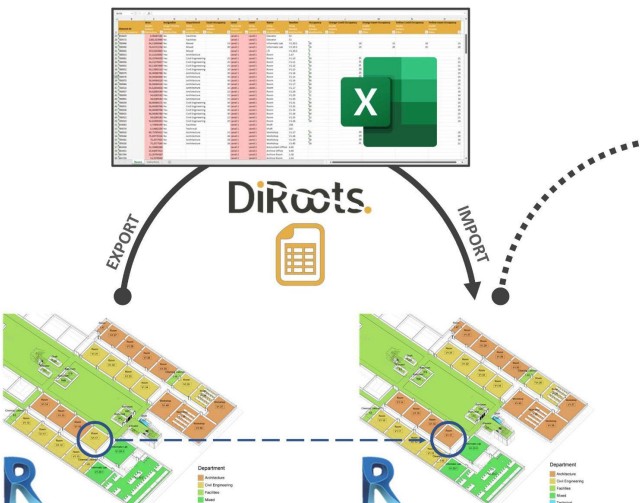

**Figure 7.** Process for updating occasional data in the system. The DiRoots® plug-in is used either to obtain or to update information in the Revit® building model. The example shows a change in an attribute value of a specific room, made by editing the spreadsheet.

For the inclusion of sensor data in the system, the devices were connected through a Wi-Fi network using the Microsoft Azure® platform, encompassing an IoT (Internet of Things) solution. Data were registered and visualized in the Azure platform and then exported to a Microsoft Excel® spreadsheet, which contained all the data records. For the process of updating data in the Revit model, a generic element representing the sensor format and location was created. In the spreadsheet for environmental data, a VBA script extracted a series of statistics, e.g., the last value of each parameter or their daily average, storing these values in the file. The update process is similar to that used for occasional data, allowing the BIM model to be fed with the measurements (Figure 8).

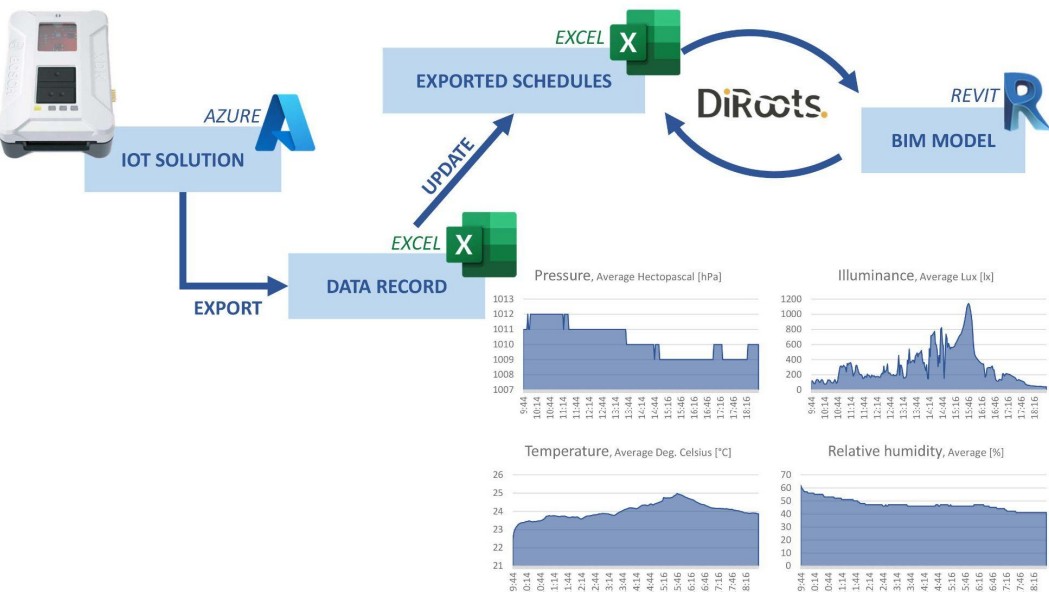

**Figure 8.** Updating of environmental data captured with sensors in the system. The DiRoots®
plug-in uses Microsoft Excel® schedules that are fed with sensor data imported from the Microsoft
Azure® platform.

To extract information from the system, a query interface (Figure 9) was developed.
This consisted of a dashboard developed in the Microsoft Power BI® interactive data
visualization software, introduced to synthesize and display summary reports for the
stakeholders. Power BI® displays interactive filters associated with graphic representations
of the building in 2D or 3D, customized tables, and interactive charts, allowing key infor-
mation to be accessed quickly and intuitively. Importing of data was performed through
the Power BI® plug-in Tracer [35]. The interface in Power BI is used for consultation,
containing read-only data. Figure 8 displays the general data flow and visualization of the
BIM-based solution.

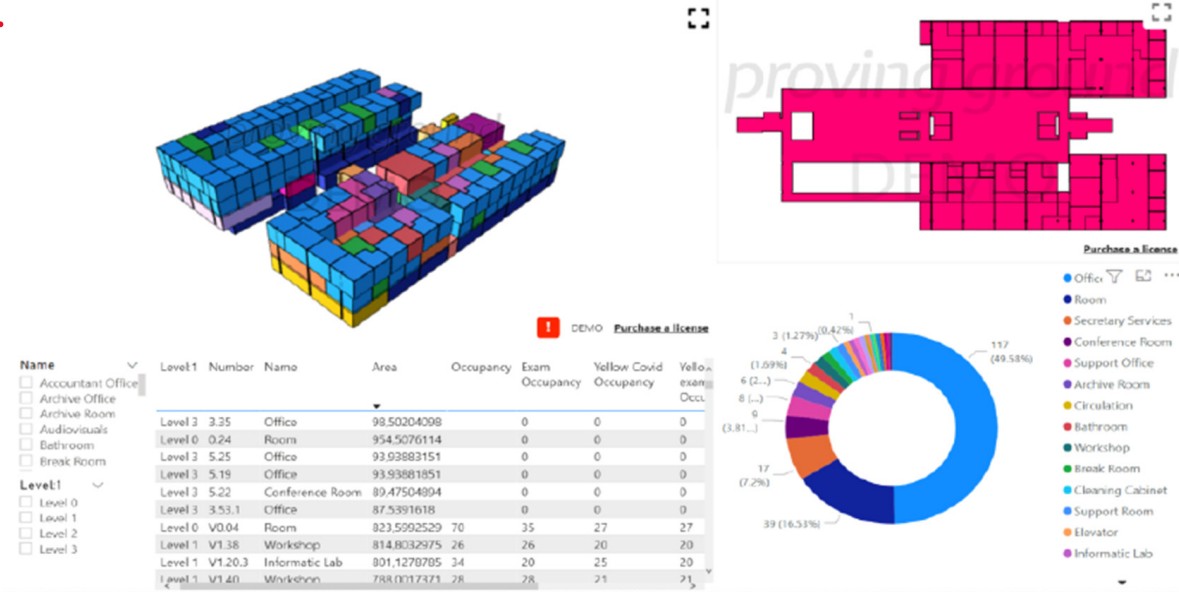

**Figure 9.** Example of a Microsoft Power BI® report for the case study building.

### 4.4. Model Data Feeding

Using the developed system, the manager only needs to control the information format and its quality, without the need to enter or edit information directly in the BIM model, apart from a simple data importing command. Data are transferred to an editable spreadsheet, which in turn can be imported into the BIM model via the DiRoots plug-in. A linked spreadsheet enables a full registry of the interventions to be kept and is fed via the VBA script. This information and the data from the BIM model accessed through the Tracer plug-in can be visualized in the Power BI interface (Figure 10).

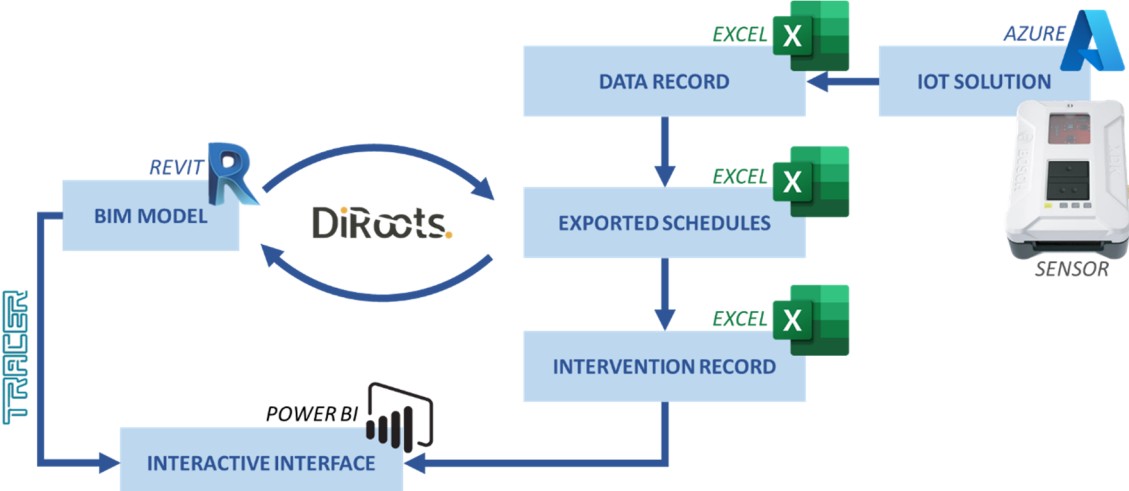

**Figure 10.** General data flow and visualization of the implemented model for occasional and continuous data.

The flow of information described above is activated through the BIM interface using a data updating tool, which changes the environmental attributes of rooms using the data stored in the associated databases linked to the sensors. Sensor data input is programmed to register the environmental conditions, recording illuminance (lx), pressure (hPa), relative humidity (%), and temperature (°C), every two minutes. The spreadsheet that stores these values can be used to extract statistical values that are then updated in the building model using the described method.

For occasional data, if, for instance, the "department info" attribute value area changes because a particular room has changed its usage, that room must be identified in the attributes list, and the change must be saved and exported using the DiRoots plug-in, updating the BIM database. If needed, the visual interface can also be updated by exporting the updated BIM database (JSON) and loading it into Power BI. Thus, the BIM database and the visual interface will remain updated.

### 5. Results

The presented prototype enabled the recording of meaningful data for building management using a BIM-centred platform with external data repositories. Dynamic data with application to building management include frequently changing values from environmental sensors and occasional updates concerning interventions in the building. In the case presented here, it was possible to integrate external Excel spreadsheets that contained both types of data, recorded automatically by sensors and provided by facility management technicians, respectively. Available plug-ins were used to enable the data flows between the data repositories and to automate the feeding, updating, and accessing of data within the centralized BIM platform.

The developed solution is accessible and can be updated by non-specialists, and a model is maintained with the most recent information for each room. In parallel, historic

records are maintained and linked to the model, allowing the retrieval of data to support interventions and decisions.

## 6. Discussion

The management of buildings using BIM software requires specialized knowledge, while the technical specialities involved in the building operation and management involve non-BIM specialists. Therefore, for the purpose of model usefulness, it is critical that accessing and updating the model is possible through user-friendly tools connected to the BIM model. There is a clear benefit to the implementation of a bidirectional data flow via spreadsheets and an interactive dashboard, facilitating the model input and output with simple interfaces.

The proposed information exchange and update model could be replicable in other applications, provided the BIM data structure reflects the tree of attributes defined in the new context. The flexibility of the BIM software used, allowing the creation of database structures linked to its geometric elements, enhances this adaptation.

In terms of geometry, using BIM it is possible to define distinct levels of spatial granularity, from a point (e.g., defined by coordinates in a wall/room) to rooms defined by walls, floors, and other elements. This favours the association of accurate positional data with the model, for instance, when it is required to locate a sensor at a particular point rather than generically in a room or on a wall. However, intricate spatial situations still present significant challenges to the modeller and must be solved through the elaboration of a more complex attribute data structure. As an example, if a room has two internal areas with different uses, it is necessary to reflect this either by expanding the tree of attributes or by using inventive geometric artefacts, such as splitting the space into two rooms without a dividing wall.

Data related to facility management needs are case-specific, and information models to maintain the system's functionality must adapt to individual space-use requirements. This is particularly true in complex structures such as multipurpose university buildings, which display a high heterogeneity of data and distinct space uses and functions.

Environmental comfort and energy efficiency are important issues in building management, for which sensor-data records support decision-making. Centring data into a BIM solution enables the data concerning spatial orientation, dimensions, and use to be understood, e.g., relating high values of illuminance to solar exposure, while supporting common data-extraction tasks such as the production of sections, plans, or thematic maps efficiently.

## 7. Conclusions

A prototype of a BIM-based system that allowed the dynamic exchange and updating of building data was developed. Despite the generic nature of BIM software, it is necessary to integrate other software into the system's architecture to develop solutions capable of handling external data sources. In the case of the developed prototype, the Tracer and DiRoots plug-ins were used to support the linkage between data sources, while Microsoft Power BI® was used to facilitate the visualization of BIM data for the technicians involved in facility management.

The most noteworthy benefit of the proposed system is the capacity to integrate information that is usually scattered into a unique platform, preventing data redundancy and enabling quick access to stored data, while providing a spatial insight via visualization.

Despite the advantages of the presented approach, some limitations can be identified, such as the need to comply with the data structures of external information, with data format requirements that must be followed by the users to enable data import into the BIM. In addition, changes in the geometry of the model, e.g., splitting or merging adjacent rooms, imply the reallocation of attributes from the altered spaces and a new definition of the linkage to the external database structure.

**Author Contributions:** Conceptualization, A.P.F. and A.B.G.; methodology, J.V.M., R.M., A.P.F. and A.B.G.; formal analysis, J.V.M. and R.M.; investigation, J.V.M., R.M., A.P.F., A.B.G. and R.B.; writing—original draft preparation, A.P.F., A.B.G. and R.B.; writing—review and editing, A.P.F., A.B.G. and R.B.; visualization, J.V.M. and R.M.; supervision, A.P.F., A.B.G. and R.B.; project administration, A.P.F., A.B.G. and R.B.; funding acquisition, A.P.F., A.B.G. and R.B. All authors have read and agreed to the published version of the manuscript.

**Funding:** This research was funded by Fundação para a Ciência e a Tecnologia, grant No. 2020.09705.BD.

**Acknowledgments:** We thank the Department of Civil Engineering, Architecture and Georesources and the manager of the Civil Engineering building of Instituto Superior Técnico, Pedro Sanches, who supported the development of the model and provided all data needed for its implementation. We also thank the anonymous reviewers for their thorough analysis and valuable comments on the manuscript.

**Conflicts of Interest:** The authors declare no conflict of interest.

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
