# Peer review of "Dynamic Data Feeding into BIM for Facility Management: A Prototype Application to a University Building"

_buildings, doi:10.3390/buildings12050645_

Round 1

Reviewer 1 Report

The authors developed a BIM-FM prototype to support operations and access updated environmental data for a university building. However, the current paper should be majorly revised before publication.

  • The abstract should be re-written since it describes lots of background, rather than contribution, novelty, key outcome, the problem statement, or purpose of the study.
  • The authors need to reorder the paper as follows:

Section 2, background and related works, the current literature review version is not comprehensive.

Section 3 should focus on the methodology.

Section 4 should be a case study

Section 5 should be a discussion, where the limitations of the proposed method should be stated, and potential solutions should be mentioned here.

Section 6 should be the conclusion

Author Response

Thank you for your comments.

The abstract was rewritten to include a better description of the novelty of the proposed methodology. We did not find in the abstract any description of background.

Section 2 (background) was organized to include a literature review on the topic of BIM-FM in general, stating the challenges of their integration, followed by a sub-section dedicated to BIM-FM for university buildings, so as to support the definition of the proposed methodology.

The structure of the paper was modified and now section 3 is only focused in the general methodology, while section 4 describes the case study.

Section 5 includes the discussion and conclusions. The limitations of the proposed methodology are presented in the last paragraph.

Reviewer 2 Report

The paper could be interesting but it needs a major revision. The main issues follow:

  • Authors not illustrate why their research is important. What is the knowledge gap they intend to overcome? What are the research questions they intend to answer? In summary, what is the contribution made by the authors' research to the scientific community?
  • The main issues are related to the method, results and discussion. In a scientific paper first it is necessary presenting the methodology, and then the case study, validating the first by applying it to the case study. The case study cannot be included in the methodology chapter.
  • The methodology that should be support by relevant literature
  • In the illustration of the research, the reference to the phases defined in the methodological approach should be made explicit (Fig. 1).
  • The results section is missing.
  • The discussion is really poor and cannot be combined with conclusions.

Other elements:

  • From page 7 the article is no longer formatted
  • Related to the sentence “As for the integration of BIM and the Internet of Things (IoT), research is still in an initial stage where most studies are mainly theoretical and conceptual (Tang et al., 2019) or oriented to prototype solutions (Desogus et al., 2021).” Indeed many others could be cited related to FM and not only. For example:

10.5194/isprs-archives-XLVI-5-W1-2022-251-2022

https://doi.org/10.5194/isprs-archives-XLII-2-W11-707-2019

10.1061/(ASCE)CO.1943-7862.0002229

10.1016/j.wasman.2022.02.027

Author Response

Thank you for your comments.

The abstract was rewritten to include a better description of the novelty of the proposed methodology. We did not find in the abstract any description of background.

Section 2 (background) was organized to include a literature review on the topic of BIM-FM in general, stating the challenges of their integration, followed by a sub-section dedicated to BIM-FM for university buildings, so as to support the definition of the proposed methodology.

The structure of the paper was modified and now section 3 is only focused in the general methodology, while section 4 describes the case study.

Section 5 includes the discussion and conclusions. The limitations of the proposed methodology are presented in the last paragraph.

Figure 1 displays the four stages of the workflow proposed for BIM-FM implementation and the names of the stages correspond to the description detailed in the sequence.

We did not understand the reference to an unformatted document after page 7.

The reference to IoT-BIM integration was rephrased. It is not our goal to explore in detail the wide field linking IoT technologies and BIM; indeed, our intention here was to refer to a state of the art where that integration is more oriented to the concepts of both technologies or to the development of exploratory case studies, from which particular improvements can be presented. This last line of research is the one that we chose: we presented a methodology - and its application in a case study - for the automation of the information update cycle for two types of dynamic data.

Round 2

Reviewer 1 Report

Great, thank you for the explanation. 

Author Response

The authors thank to the comments of the reviewer.

Reviewer 2 Report

Few changes are highlighted in the submitted file so it is not easy to identify changes and improvements in the article. However, some changes have been made but many important points are still missing for the article to be scientific. The research gap has not been indicated in the paper. There is an explanation in the letter to the reviewer but this should be included in the main text, to identify the gap.

There is also no results paragraph in the article. The website buildings state "all manuscripts must contain the required sections: Author Information, Abstract, Keywords, Introduction, Materials & Methods, Results, Conclusions, Figures and Tables with Captions, Funding Information, Author Contributions, Conflict of Interest and other Ethics Statements. Check the Journal Instructions for Authors for more details.". Again, as already indicated in the first round by the reviewer, The discussion is really poor and cannot be combined with conclusions. The bibliography has not been improved as suggested by the reviewer, only the sentence has been changed in a convenient way.

Author Response

Thank you for your comments, which we hope to have addressed correctly in the new version of the manuscript.

The research gap is now identified in the Introduction. We focused on the development of a solution that addresses the update of dynamic data from diverse sources into a BIM oriented for the facility management. This is also reflected in the title given to the article.

The structure of the manuscript was modified to accommodate the suggestion of splitting Conclusions and Results into separate sections, and improve their contents and clarit.

As of additional bibliographic references, we chose to include those that concentrate on IoT /BIM integration from both the conceptual and prototype-oriented perspectives.

Round 3

Reviewer 2 Report

Referee suggest to support the conclusions by the results.

Author Response

We tank the reviewer comments. In order to answer to the review request the structure of the sections at the end of the manuscript has been changed (a new section entitled "Discussion" has been added), and their content, have been adapted and expanded to reflect the reviewer's observations.